# CytoSIP: an annotated structural atlas for interactions involving cytokines or cytokine receptors
Lu Wang [1,2,3,7], Fang Sun[1,4,7], Qianying Li[1], Haojie Ma[1], Juanhong Zhong[1], Huihui Zhang [1], Siyi Cheng[2,3], Hao Wu[2,3], Yanmin Zhao[1], Nasui Wang [5], Zhongqiu Xie [6], Mingyi Zhao[4] ✉, Ping Zhu[2,3] ✉ & Heping Zheng [1] ✉

Therapeutic agents targeting cytokine-cytokine receptor (CK-CKR) interactions lead to the disruption in cellular signaling and are effective in treating many diseases including tumors. However, a lack of universal and quick access to annotated structural surface regions on CK/CKR has limited the progress of a structure-driven approach in developing targeted macromolecular drugs and precision medicine therapeutics. Herein we develop CytoSIP (Single nucleotide polymorphisms (SNPs), Interface, and Phenotype), a rich internet application based on a database of atomic interactions around hotspots in experimentally determined CK/CKR structural complexes. CytoSIP contains: (1) SNPs on CK/CKR; (2) interactions involving CK/CKR domains, including CK/CKR interfaces, oligomeric interfaces, epitopes, or other drug targeting surfaces; and (3) diseases and phenotypes associated with CK/CKR or SNPs. The database framework introduces a unique tri-level SIP data model to bridge genetic variants (atomic level) to disease phenotypes (organism level) using protein structure (complexes) as an underlying framework (molecule level). Customized screening tools are implemented to retrieve relevant CK/CKR subset, which reduces the time and resources needed to interrogate large datasets involving CK/CKR surface hotspots and associated pathologies. CytoSIP portal is publicly accessible at https://CytoSIP.biocloud.top, facilitating the panoramic investigation of the context-dependent crosstalk between CK/CKR and the development of targeted therapeutic agents.

Cytokines (referred to as CK hereinafter) include growth factors and are a large family of mainly soluble proteins and glycoproteins that are molecular entities functionally act as an extracellular signal to critically modulate the immune system[1]. These small signaling molecules (typically between 5 kDa and 25 kDa) are secreted primarily by leukocytes but can also be produced by other cell types, such as endothelial cells, epithelial cells, and fibroblasts. CKs exert a vast series of immune regulatory actions important to human biology and could alter the normal physiological function to disease via

cytokine receptors or growth factor receptors (referred to as CKR hereinafter) on cell surface[2]. Signals conveyed by CKs are essential for the generation, survival, and homeostasis of immune cells, as well as for the generation of immune responses upon external stimuli[3]. Nevertheless, the resulting translation of extracellular signal to intracellular signal transduction and subsequent amplification could also cause potentially harmful or even life-threatening pathologies[4]. The overproduction of pro-inflammatory CKs that leads to acute respiratory distress and extensive

[1]Bioinformatics Center, Hunan University College of Biology, Changsha, Hunan 410082, China. [2]Guangdong Cardiovascular Institute, Guangdong Provincial People's Hospital (Guangdong Academy of Medical Sciences), Southern Medical University, Guangzhou, Guangdong 510100, China. [3]Guangdong Provincial Key Laboratory of Pathogenesis, Targeted Prevention and Treatment of Heart Disease, Guangzhou Key Laboratory of Cardiac Pathogenesis and Prevention, Guangzhou, Guangdong 510100, China. [4]Department of Pediatrics, The Third Xiangya Hospital, Central South University, Changsha, Hunan 410006, China. [5]Division of Endocrinology and Metabolism, The First Affiliated Hospital of Shantou University Medical College, No. 57 Changping Road, Shantou 515041, China. [6]Department of Pathology, School of Medicine, University of Virginia, Charlottesville, VA 22908, USA. [7]These authors contributed equally: Lu Wang, Fang Sun. ✉e-mail: mingyi@csu.edu.cn; tanganqier@163.com; hz5p@hnu.edu.cn

**Table. 1 | Data statistics in CytoSIP**

| Number of occurrence | | CK | | CKR | |
|---|---|---|---|---|---|
| Key features in CytoSIP | | Seq.ID. ≥95% | Seq.ID. <95% | Seq.ID. ≥95% | Seq.ID. <95% |
| SNP | | 2853 | | 2982 | |
| Interaction interfaces | (2a) CK-CKR, CK-CK, CKR-CKR interaction interfaces | 48109 | | 63,004 | |
| | (2b) Epitopes | 1537 | 10,940 | 1101 | 18,873 |
| | (2c) Targeted surfaces — Small molecules | 2359 | 35,737 | 3726 | 107,026 |
| | Metals | 115 | 1719 | 122 | 2711 |
| Phenotypes | | 7167 | | 4746 | |

Identity refers to the similarity to the CK or CKR structure we selected. The CK-CKR interaction interface uses only structures with sequence identity >95%, while the epitopes, targeted surface for small molecules and metals also use structures with sequence identity <95%. SNPs in CytoSIP consist of missense mutations. Epitopes on CK or CKR feature interactions with antibodies such as monoclonal antibodies or nanobodies. Targeted surfaces on CK or CKR feature interactions with small molecules or drugs. Phenotype data include diseases that are known to associate with at least one CK or CKR.

tissue damage is known as cytokine storm and has been the culprit in many fatal viral infections[5]. Similar inflammatory cytokine storm is known to be responsible for pathologies such as sepsis, main cause of hospital-based mortality among the patients[6]. Cytokine release syndrome also causes severe side effects in cancer treatments, such as immunotherapies[7] and chimeric antigen receptor T-cell therapies[8]. Similarly to the disconnect between the severity of immune-related symptoms and disease outcomes in infectious disease, patients without cytokine release syndrome may also experience tumor growth and, therefore, production of CKs could lead to complete tumor remission[9]. These observations imply that appropriate modulation of CK signaling must be realized for the benefits of immune cell-based therapies to either attain physiological CK signaling or extensively suppress pathological effects[10].

The desired immunotherapeutic effects of natural CKs are often mitigated by toxicity or inefficacy, resulting from CKR pleiotropy and undesired activation of off-target cells and dysfunction of organ. With the deepening of our understanding of the structural principles governing the molecular level-specific CK-CKR interactions, mechanism-based manipulation of CK signaling through protein engineering becomes increasingly feasible. Modified CKs, both agonists and antagonists, have been engineered with defined target cell specificities, providing mechanistic insights into cytokine biology and signaling[11]. Therapeutic agents targeting CKs[12] include antibodies against Tumor Necrosis Factors α (TNFα)[13], TNF11[14], Erythropoietin, or Granulocyte-Colony Stimulating Factor (G-CSF)[15]. Those targeting CKRs include antibodies against Interleukin-6 Receptor (IL6R)[16], IL9R[17], IL13R[18] or IL17R[19]. The development of these therapeutics has validated the initial promises in the field of cytokine biology[20].

Identifying drugs that modulate specific disease-related CK-CKR interactions is a promising strategy for drug discovery[21]. However, the design of drugs to disrupt such protein–protein interactions (PPIs) is challenging. For chemical therapeutic agents, many potential drug-binding sites at PPI interfaces are "cryptic": unoccupied cryptic sites are often flat and featureless and thus not readily recognizable in crystal structures, with the geometric and chemical characteristics of typical small-molecule binding sites only emerging upon ligand binding in ternary complex[22]. For antibody drugs, the adaptive immune response produces an epitope-diverse, polyclonal antibody mixture capable of neutralizing invading exogenous molecules through binding interference[23]. Therefore, precise and in-depth knowledge of these "antigenic clusters" is critical for developing diagnostics and therapeutics targeting infectious, allergic, and autoimmune diseases and carcinoma[24].

Drugs targeting CK or CKR may not always be effective due to the presence of genetic variance in population, which needs to be addressed by precision personalized medicine (PPM)[25]. Naturally occurring mutations of various CK or CKR have been identified in patients[26] as missense Single nucleotide polymorphisms (SNPs). SNPs are caused by changes in gene expression, protein stability, localization, and function; and are causally related to the phenotypic characteristics of individuals. While most SNPs exist in non-coding sequences and influence mRNA expression, splicing, and stability, some SNPs do cause mutations in coding regions and are referred to as missense SNPs. Most missense SNPs have not been functionally characterized, most likely due to their discovery in more random genome sequencing methods rather than through underlying disease-driven sequencing strategies[27].

Therefore, unraveling molecular interactions at the interface between CK and CKR and resulting mechanism for signal transduction would provide a targeted strategy in the rational design and engineering of therapeutics. While a large part of the PPIs data considers protein molecules as the minimal unit[28], refining the perspective from full-length protein chains to domain–domain interactions or even targetable surface regions within protein crystals extends our ability to generate specific hypotheses about the functional interactions between CK and CKR[29]. However, to the best of our knowledge no such computational simulation protocol is available to enable prompt universal access to annotated structural surface regions on CK or CKR. Analyzing and evaluating such information often requires examining hundreds or even thousands of available protein structures in the Protein Data Bank (PDB)[30], which is tedious and challenging even for scientists specifically trained in structural biology or bioinformatics. A database of CK-CKR interactions and surface hotspots featuring antigenic epitopes and binding pockets harboring small-molecule compounds specifically on CK-CKR complexes would provide a foundational framework for the design of candidate drugs targeting the CK-CKR signaling pathway.

Several excellent databases have been developed that focus on CK immune reactions and vaccine design[31], CK-CKR interactions[32], or CKs target genes[10]. There are also databases reporting transcriptional regulation for CKs in human and mouse[33]. Related studies also feature protocol to predict functionally related SNPs through in-depth structural guidance analysis[34]. Despite these advances, there is a lack of integrated and annotated knowledgebase documenting structural features of the surface regions and intermolecular interfaces on CK-CKR structure complexes. Herein, we report the development of CytoSIP (Single nucleotide polymorphisms (SNPs), Interface, and Phenotype), a structural database of CK and CKR encompassing SNPs, PPIs, and diseases associated with CK or CKR in a single resource. PPIs considered in CytoSIP include CK-CKR interaction interface, drug-binding sites information, epitopes, and other interfaces between CK-CKR complexes and oligomers. CytoSIP creates an atlas upon which one can annotate newly characterized functional sites. Our database provides one-stop, combinatory representations and annotations of hotspots on the structural proteomes of CK or CKR. CytoSIP is a fundamental resource for the rational design of preventative, diagnostic, and therapeutic procedures targeting CK or CKR and is publicly accessible at https://CytoSIP.biocloud.top/.

## Results
### Content of the CytoSIP database
CytoSIP database is an online resource of comprehensive data about CK and CKR, containing a rich and intercorrelated annotation on genetic variants, structural interfaces, disease phenotypes, and hotspots on CK or CKR (Table 1). The central architecture of CytoSIP includes the following key features: (1) SNPs on CK or CKR; (2) interactions involving CK or CKR at the domain level, including CK-CKR interfaces, oligomeric interfaces, epitopes, or other drug targeting surfaces; and (3) diseases and phenotypes associated with CK/CKR or SNPs.

Interactions involving CK or CKR form the foundational framework for all other annotations in CytoSIP and can be further subdivided into three categories: 2a) the interaction interface between CK and CKR, or the oligomeric interaction interface between two CKs or two CKRs; 2b) the CK or CKR epitope targeted by monoclonal antibodies or nanobodies; and 2c) the

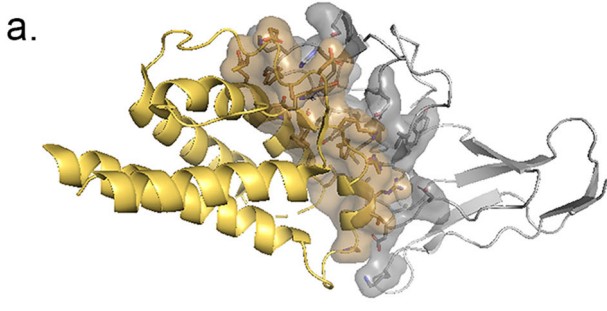

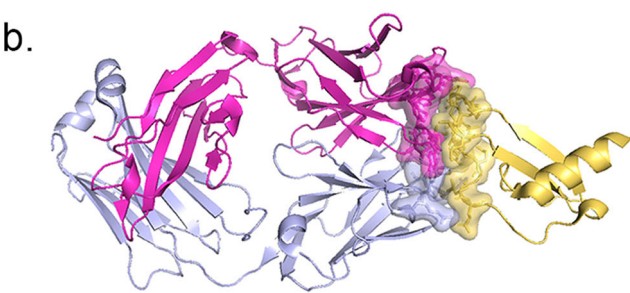

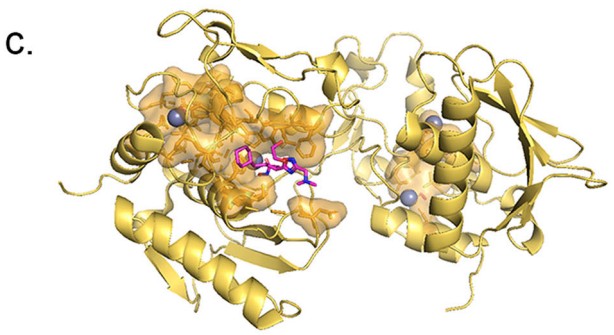

**Fig. 1 | Interfaces in CytoSIP. CKs are colored in yellow, and CKRs are colored in light gray.** Heavy chain and Light chain of antibody are colored in magenta and lavender accordingly. The interfaces show sticks and surface on the structure. This coloring scheme applies to all protein–protein interface figures unless otherwise stated. **a** Category 2a interaction interface in CK-CKR complex (IL2-IL2R complex, PDB ID: 1Z92); **b** Category 2b Epitope in CK-antibody complex (CXC motif chemokine 2 complexed with its antibody, PDB ID:5OB5); **c** Category 2c Targeted surface in CK-drug complex (BMP2 complexed with L-tartaric acid, PDB ID: 4UI2).

targeted surface on CK or CKR, which is the binding site of small-molecule drugs, polypeptide drugs, or metal drugs. CytoSIP compiles all known interaction interfaces from CK-CKR pairs (Fig. 1a), including 38 CKR dimers and 3 CKR trimers. From 2919 targeted antibodies, we get the epitopes (Fig. 1b). And from 2360 drugs, 2238 of which were approved by the FDA, we get the targeted surface (Fig. 1c).

## Interactions involving CK or CKR in CytoSIP

CytoSIP catalogs three types of category 2a interaction interfaces: CK-CKR interaction interface, CK-CK interaction interface, and CKR-CKR interaction interface. While most structural data refers to CK-CKR interaction interface, their interaction is not exclusive, since a CK can interact with two or three CKRs using different surfaces or interact with a heterodimer composed of two different CKRs. For example, Interleukin-4 (IL4), a CK critical to developing T-cell-mediated humoral immune responses associated with allergy and asthma, can exert its actions through different receptors. In our database, IL4 interaction with IL4RA, IL2RG (Fig. 2a),

IL13RA (Fig. 2b), IL13RB and heterodimer IL13RB/IL4RA, IL2RG/IL4RA, IL13RA/IL4RA. The receptor IL4RA interacts with IL13 (Fig. 2c) besides IL4.

For category 2b epitopes, CytoSIP identifies all antibody-antigen interfaces in which CK or CKR are antigens. Antibodies with different epitopes may exhibit different effects. For example, CytoSIP identifies three specific antibodies that target Interleukin-6 (IL6): olokizumab[35], 61H7[36], and 68F2[36]. The representative structure of IL6 in the CytoSIP reveals the relative locations of three different epitopes (Fig. 3a–c) that antagonize the CK-CKR interaction interface (Fig. 3d–f). Insights into the mechanism of neutralization by the three antibodies can explain the distinct potency of these antibodies against IL6, a CK involved in metastasis, tissue invasion, and mediating inflammatory reactions in autoimmune diseases and cancer. Neutralizing antibodies against IL6 and its receptor have been approved for therapeutic intervention or are in advanced stages of clinical development[36].

## SNPs on the interface in CytoSIP

SNPs of CK or CKR obtained from dbSNP database include intronic variations, in-frame deletions, insertions, initiator codon variants, missense variants, non-coding transcript variants, and synonymous variants. CytoSIP surveys only missense mutations that cause an amino acid change in the CK or CKR coding sequence. SNP on CK or CKR, directly aligns dbSNP information to DNA and protein sequence information to illustrate a genetic SNP landscape exemplified for all CK/CKR and subsequent diseases linkage of SNPs.

Genetic variants that cause missense mutations at the category 2a CK-CKR interface may suggest information deserving further attention. Taking TNFA as an example, 16 SNPs are found on the interaction interface of TNFA-TNFR1A and TNFA-TNFR1B. These include rs1800620 polymorphism[37] with Helicobacter pylori in Type 2 Diabetes Mellitus (Fig. 4a), and rs4645843[38] related to symptom burden and QOL outcomes in lung cancer survivors (Fig. 4a).

The information on SNPs attributed to the important role of genetic variations in protein interactions has been used to formulate innovative therapeutic strategies targeting category 2b epitopes. For example, IL6R genetic variants enable the investigation of IL6R inhibitors including tocilizumab and sarilumab for drug repurposing to treat COVID-19[39]. Another example features the use of gevokizumab, which has desirable pharmacokinetic properties to inhibit IL-1β-specific activity, which is envisioned to be a better strategy than blocking the activating receptor. The crystal structure of the IL-1β:gevokizumab complex (Fig. 4b) validates the published mutagenesis studies that identified IL-1β residues E96, K97, and Q116 (rs1379621278)[40,41] as being critical for antigen binding (Fig. 4c).

## Disease-associated interface in CytoSIP

CytoSIP provides 8069 human diseases correlated with CK/CKR or their variants, including 7167 disease-CK associations and 4746 disease-CKR associations (Table 1). This information helps researchers explore the molecular basis of human diseases and characterize and validate disease-related candidate genes. Combined with our structural data, CytoSIP aids in quickly identifying interfaces that may be related to diseases, such as a CK-CKR pathway associated with a disease and drugs targeting the CK-CKR interface. CytoSIP also provides SNPs on the interface to help researchers analyze the effect of genetic variants on targets associated with the physiological or pharmaceutical processes (Fig. 5). For example, 39 CKs, 13 CKRs, and 6 SNPs correlate with Acute Coronary Syndrome, including six CK-CKR pairs and two SNPs on the CK. One such CK-CKR pair is the interaction interface between IL2 and IL2RA, which may be targeted for better treatment of Acute Coronary Syndrome.

## The CytoSIP web portal

On the CytoSIP website, there are three types of inputs: (i) gene; (ii) protein; (iii) disease. User can browse CK-CKR pairs, hotspots on protein structure, disease-related CK/CKR, and SNP (Fig. 6). Except for Disease

**Fig. 2 | CK-CKR Interaction interfaces in CytoSIP.** IL4 interaction with three different CKRs. CKs are colored in yellow, and CKRs are colored in light gray. All the interactions of each CK or CKR can be found on the corresponding page in CytoSIP. **a** Interaction interface of IL4 with IL4RA, IL2RG (PDB ID: 3BQL). **b** Interaction interface of IL4 with IL13RA (PDB ID: 3BQN). **c** Interaction interface of IL4RA with IL13 (PDB ID: 3BPO).

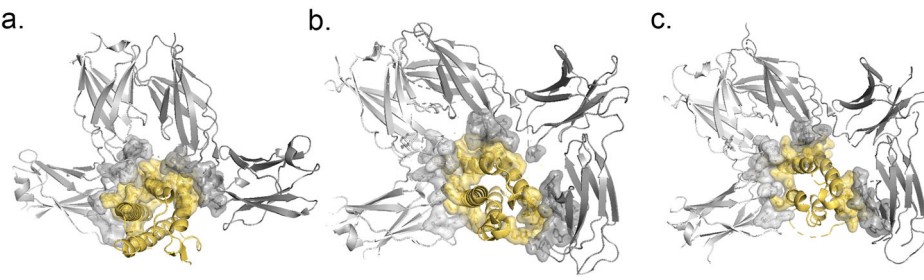

name input, all inputs result in a list of PDB structures, providing a comprehensive overview of homologous structures for any given query. Users can check the clusters from one structure for chain–chain, domain–domain, and domain–peptide interactions. The datasets are either downloaded from the publicly accessible server or produced in-house. Each CK or CKR is analyzed through a standardized workflow, with intermolecular interfaces and hotspots identified, filtered, and imported into the database using a streamlined bioinformatics analysis pipeline specified in detail below.

The CytoSIP web interface provides a panoramic view for each CK or CKR structure (Fig. 7). The database selects the Uniprot sequence as the reference sequence for standard amino acid positions other sequences can align to, and a representative structural model as the framework other annotations can map upon. Domains and structures are annotated on the unified sequence, while interaction interfaces, SNPs, and all hotspots are annotated on both the unified sequence and the surface of the representative structure to allow for comparative and correlative analysis. These utilities allow for quick comparative analysis and visualization of interaction interfaces based on experimentally determined complex structures.

## Discussion

The Protein Data Bank holds a tremendous amount of information on the interactions of molecules in biological systems, but it can be challenging to access and analyze rapidly. CytoSIP is designed to accomplish this task with a unique combination of features that enable users to obtain the rich structural information available for CKRs quickly and easily. For all protein interactions, CytoSIP provides the coordinates of the complexes and visualizes the interactions for chain–chain, and domain-domain interaction complexes as a framework to perform in-depth feature analysis of the complexes. While some of these features are individually or jointly available in other databases, analyzing them in the structural context allows us to access the full range of currently available structural information.

The clustering of interactions is crucial for providing evidence in favor of the biological relevance of any specific interaction analyzed in CytoSIP. Some protein families have evolved different ways of forming oligomers in various branches of their phylogenetic trees. These interactions will appear in distinct CytoSIP complexes with non-overlapping protein sets. In other cases, some proteins form larger oligomers that contain two or more distinct interfaces. Some proteins exist in different oligomers under different conditions with different functional properties[42]. CytoSIP helps to sort out which PDB entries contain which interfaces of these oligomers. For example, the CKR pleiotropy of IL4/IL13 (Fig. 2) illustrates how different CKs can induce divergent signaling responses through shared receptors in different cellular contexts[43]. The presence of CKR sharing further suggests that the extracellular CK-CKR interactions can influence the potency of receptor signaling while intercellular mechanisms are in place to dictate downstream signaling specificity. By contrast, other online resources available to date only lump all interactions of domains into one group and do not distinguish between different binding modes.

CytoSIP has some limitations based on the data currently available in the PDB. For some CK-CKR complexes, there may be no crystal form (even if there are multiple entries), and thus, there will be no clusterable interfaces

across crystal forms. Some proteins are poorly annotated by Pfam, missing some N or C terminal elements of secondary structure that may form a large portion of a homodimer or heterodimer interface. This compromises our ability to cluster some domain–domain interfaces, which is partly compensated by continuing to provide the chain–chain clustering since these include the entirety of each chain. The paradigm of CKR signaling is through ligand-mediated dimerization or ligand-mediated reorganization of preformed receptor dimers. The interaction interface of dimers is not included in the database because their structures were poorly defined, and the related publications currently lack quality control information on their definition and structure. CytoSIP will become increasingly valuable as more biological data on diverse CKs and CKRs becomes available.

CytoSIP provides comprehensive annotations to the members of each complex for protein–protein interactions (at the chain and domain levels) and interactions of domains with peptides and ligands on the structural surface level. These annotations include PDB ID and chain, crystal form, UniProt identifiers, and genetic variants on each representative structure. These provide information on the biological relevance of each annotated surface area on the structures, which may be followed up in the papers describing the structures. By contrast, other online resources available to date do not contain information on the identities of proteins in the complexes they list, instead providing only PDB identifiers and links to the PDB website. Users need to interrogate through hundreds of CK structures could fail to identify critical features of their interest in the lengthy process.

Interfaces from protein complex networks can drive cellular signaling and cellular functions. The CytoSIP provides quick and annotated access to a resource of the most integrative complex structures involving CK or CKR. Precise modulation of a target interface helps explain the role of the interface in cellular events and possesses therapeutic potential[44]. Analyzing the interface may also provide strategies to improve the specificity of drugs, while labeling other information on the interface to enrich the crystal structure annotation can open more research avenues. For example, antagonists of inflammatory factors have been developed for the treatment of various inflammatory diseases, such as Crohn's disease, rheumatoid arthritis, ankylosing spondylitis, and psoriasis[45]. The CK-CKR interaction interface may help the design of these antagonists. CK or CKR pleiotropy and/or undesired activation of off-target cells may also lead to toxicity or lack of efficacy of the immunotherapeutic. These convergent molecular and structural principles can provide more meaningful signals for developing hypotheses for biological interfaces, and should be considered when undertaking a CK engineering experiment[11].

Comprehensive and comparative analysis of different types of interactions and genetic variants may reveal valuable correlations. For example, CytoSIP characterizes that IL2 interacts with three CKRs and three heterodimers, with three complex structures and six SNPs on the CK-CKR interaction interface. Besides, one IL2 complex structure with a specific antibody and three IL2 complex structures with three different drugs are also observed (Fig. 7). Preclinical studies of the T-cell growth factor activity of IL2 resulted in this CK becoming the first immunotherapy approved ~30 years ago by the US Food and Drug Administration (FDA) to treat cancer. The knowledge about interaction interface between IL2 and its tripartite receptor has contributed to the development of cell type-selective engineered IL2 products[46]. The CK-CKR interaction interfaces and targeted surface

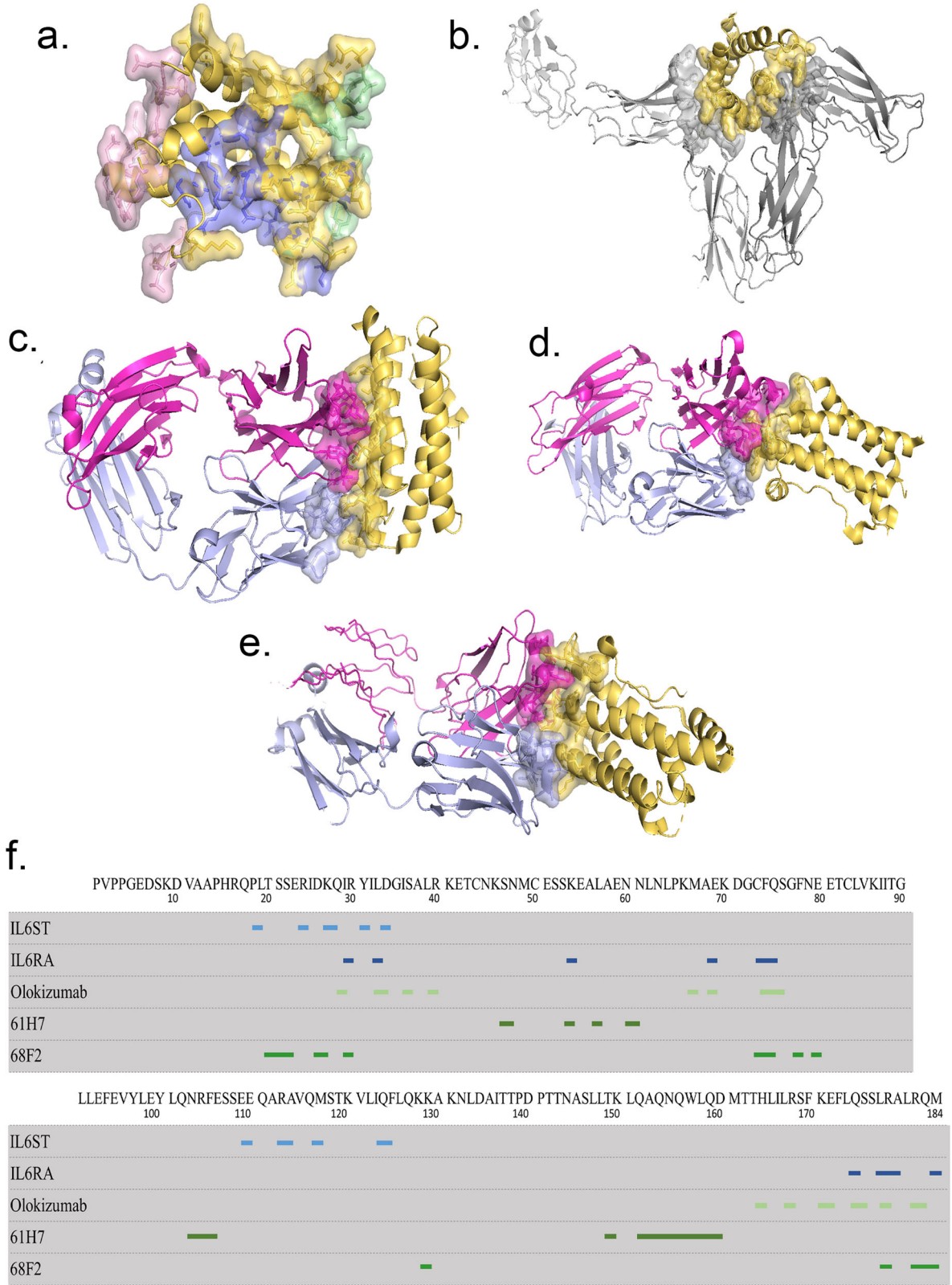

**Fig. 3 | Comparison of interfaces between IL6 and three IL6-specific antibodies targeting different IL6 epitopes.** CKs are colored in yellow and CKRs are colored in light gray. Heavy chain and Light chain of antibody are colored in magenta and lavender accordingly. **a** Three distinctive IL6-specific antibodies targeting epitopes mapped onto the IL6 representative structure: olokizumab (colored in purple), 61H7 (colored in pink) and 68F2 (colored in green). **b** Interaction interface of IL6 with IL6ST and IL6RA (PDB ID: 1P9M); **c** Epitope of IL6 with olokizumab (PDB ID: 4ZS7); **d** Epitope of IL6 with 61H7 (PDB ID: 4CNI); **e** Epitope of IL6 with 68F2 (PDB ID: 4O9H); **f**. Human IL6 sequence is marked alongside with interaction interface of IL6ST (light blue strips), IL6RA (dark blue strips), and epitopes of olokizumab, 61H7, 68F2 (green strips).

**Fig. 4 | SNPs on interface in CytoSIP.** CKs are colored in yellow and CKRs are colored in light gray. Heavy chain and Light chain of antibody are colored in magenta and lavender accordingly. Missense SNPs are labeled and colored in red on the CK-CKR interface. **a** The 16 SNPs found on the interaction interface of TNFA-TNFR1A or TNFA-TNFR1B mapped onto a representative TNFA structure (PDB ID: 7KPB); **b** The structure of IL-1β in complexed with gevokizumab with epitope and paratope shown in surface representation (PDB ID: 4G6M); **c** Mutants E96A, K97A, and Q116E marked on the epitopes of IL-1β with gevokizumab (PDB ID: 4G6M).

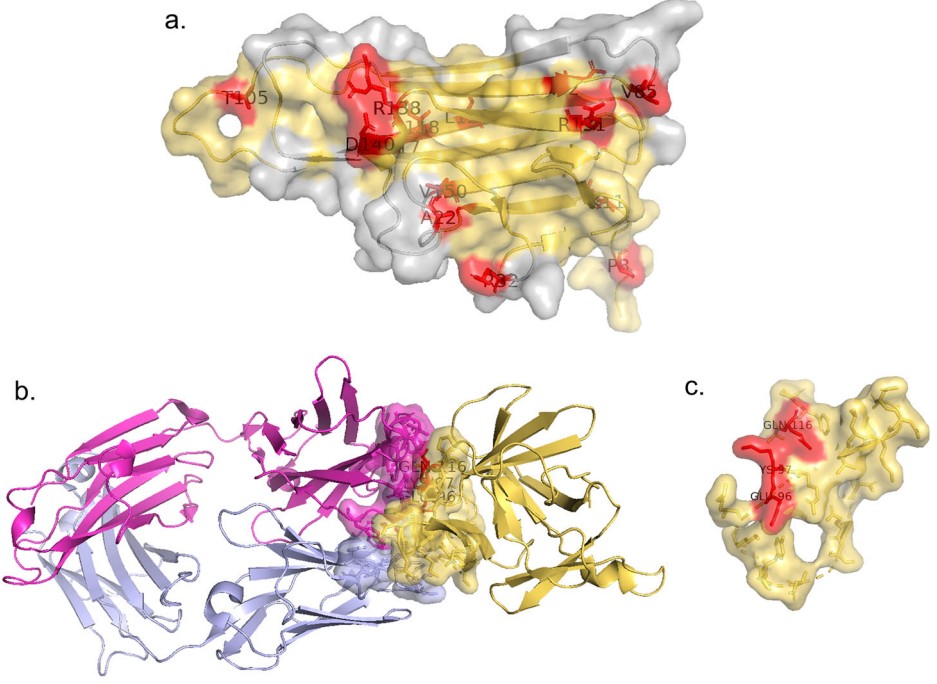

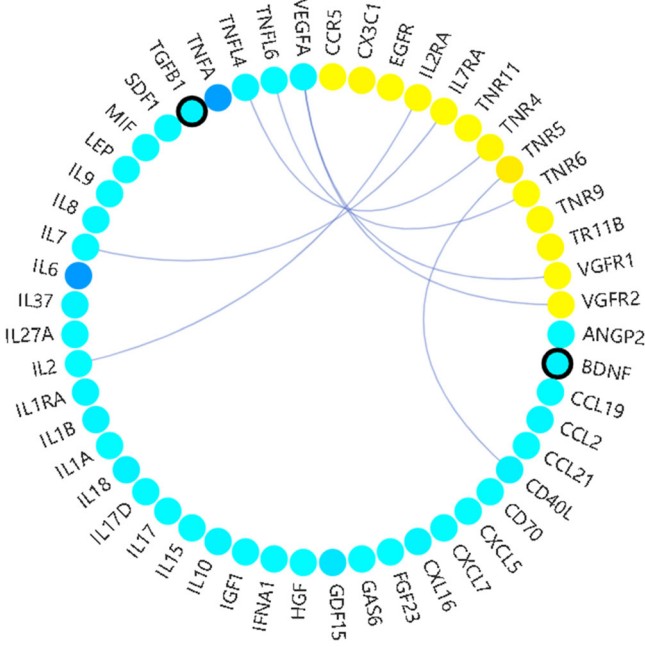

annotated for IL2 structure herein should provide a wealth of correlated information in a well-organized format to further advance IL2 research.

Evidence suggests that a subgroup of patients with severe COVID-19 might have cytokine storm syndrome[47]. Moreover, the blockade of some CKs has been used to manage cytokine release syndromes following CAR-T-cell therapy in certain patients with cancer[48]. The resurgence of clinical use of CK-related treatments brings us back to the importance of CKs; however, research and drug development are tricky due to the complexity of CK binding and the critical role of SNP in it.

CytoSIP provides correlation information about disease and protein or SNP. It forms the ground of scientific hypotheses that missense SNP may lead to the change of residues, which may induce the change of the CK and CKR interaction interface and ultimately cause disease. This information is valuable for formulating therapeutic strategies to target a specific disease and conducting an in-depth analysis of the inherent biological significance. On the other hand, genetic variance also causes discrepancy in population that requires consideration in the field of precision medicine.

The basis of human disease genetics is central to drug discovery. CytoSIP provides the most comprehensive CK gene and structure-related data to date, yielding critical mechanistic insights into cytokine biology and signaling. As our understanding of the structural principles of CKR-targeted drug interactions advance, mechanism-based manipulation of signaling through protein engineering has become an increasingly feasible and robust approach. The combination of these data contained in CytoSIP can be used to compare the similarity of each interface, and the off-target effects of drugs can be effectively controlled by avoiding similar targets at a relatively early stage during the design of targeted drugs.

In summary, we constructed a universal CK and CKR structure database containing structure annotations for interaction interfaces and different types of hotspots in a unified format. The database contains 243 CKs and 138 CKRs with various lines of associated experimental and clinical evidences. We also annotated genetic variants and disease information to correlate valuable information that can be used to predict the relationship between diseases and CKs/CKRs, structures, and SNPs. We hope that with the extension of CytoSIP to CK/CKR and further enhancements described in this paper, CytoSIP can be used to understand the mechanisms of CK-mediated signaling transduction pathways in physiological, pathological, and pharmaceutical processes and to formulate therapeutic strategies to target the processes.

**Fig. 5 | Disease-related CK/CKR and SNPs in CytoSIP.** The results are shown for Acute Coronary Syndrome. CKs are colored in cyan-blue, while CKRs are colored in yellow, with CK-CKR pair indicated by linking lines. Circles filled with darker hues indicates the higher correlation with the indicated disease. Circles with a bold black border indicates the presence of SNP on the CK or CKR correlated with this disease.

## Methods
### Selection of cytokines and cytokine receptors
We collect human cytokines and growth factors (referred to as CK hereinafter), and cytokine receptors and growth factor receptors (referred to as CKR hereinafter) reported in the Kyoto Encyclopedia of Genes and Genomes (KEGG)[49]. The pairing and oligomeric states of CK and CKR are also from the KEGG pathways. Many CKR signaling functions through ligand-mediated dimerization or ligand-mediated reorganization of preformed receptor dimers. Oligomeric information is included in the database.

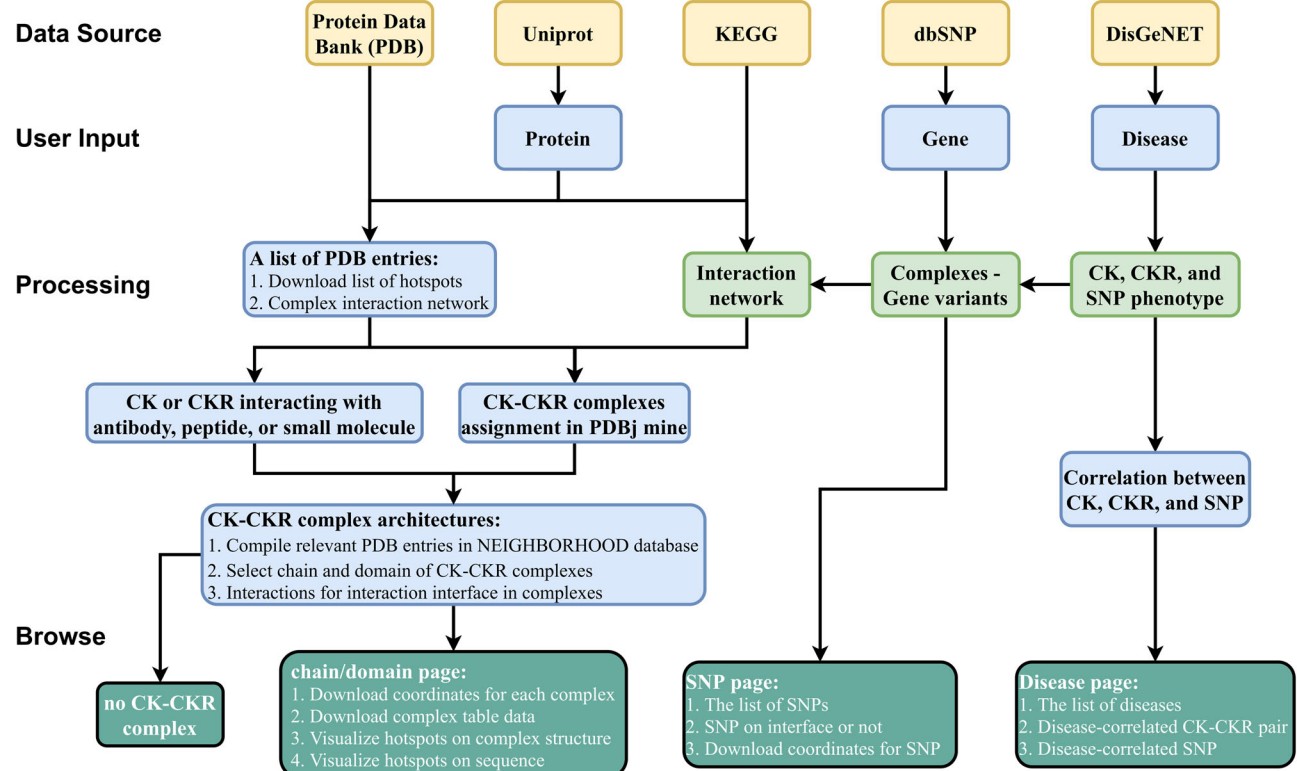

**Fig. 6 | The infrastructure of the CytoSIP website.** Three types of queries can be input by users: Protein, Gene, and Disease. The user can also browse all available gene IDs, rsIDs, CK-CKR pairs, drugs, and antibodies. A protein page provides a list of PDB structures. Clicking on one structure leads to the same page as querying by that PDB ID, which shows the complexes architecture of the entry. From the protein page, a user can select and show the details of different categories of interfaces including interaction interfaces, targeted interfaces, epitopes, metal binding sites, and SNP. The list of different interactions and complexes can be downloaded from the protein pages or the complex pages. From the disease entry page, a user can select and display disease-correlated CK/CKR and SNP and the CK-CKR pairs involving SNPs.

The gene symbols of CK or CKR from KEGG are verified to adopt HUGO Gene Nomenclature Committee (HGNC) standard[50]. We use the standard gene symbols of each CK or CKR to confirm the presence of the candidate CK or CKR proteins in UniProt[51] and to retrieve the corresponding amino acid sequences. After removing redundant records, this procedure yields a dataset of 243 CKs from 7 CK families and 138 CKRs from 13 CKR families. We assign Uniprot ID for each protein in the dataset to link with information from other databases. If a CK or CKR has two or more Uniprot IDs, both IDs are included.

### Processing of structural data
We align amino acid sequences of CK or CKR with amino acid sequences in the PDB database using FASTA36[52]. Structures containing CK or CKR chains that have a sequence similarity higher than 30% were downloaded using the 10 March 2023 version of the PDB[53], with the corresponding metadata stored in a local implementation of the PDBj Mine RDB version 2[54]. All PDB structures containing at least one CK or CKR protein component are processed using the Neighborhood database[55], a relational database that stores the interactions between the atoms and residues of all modeled protein structures. The Neighborhood database also stores other PDB-derived information, which provides an effective way to store, query, and classify the intermolecular interactions and hotspots in protein structures.

### SNPs
Genetic variants are obtained from the NCBI SNP database[56] using the Gene ID of the CK or CKR. Only missense variants that cause change in the CK or CKR amino acid sequence are retained and subject to further processing. We link the rsID of each SNP record to the corresponding GeneID and further annotate each rsID using Annovar[57]. Information of genetic variants

linked to each rsID record is further mapped and labeled to site information in the PDB structure. Changes in amino acid identity caused by missense mutation are subject to further analysis.

### CK-CKR interfaces
The interactions between receptors and ligands are characterized using the Neighborhood database and further verified for consistency with the KEGG database[49]. The sequences of the two interacting macromolecular partners from PDB complex structures are searched against the known list of CK or CKR sequences in the dataset obtained from 2.1 using FASTA36[52]. If sequence search results in a sequence similarity >95% for both interacting partners, the structural interface is used as a framework of the interaction site to exemplify the interaction relationship between CK and CKR. If sequence search results in the sequence similarity of CK between 30% and 95%, while the sequence similarity of CKR >95% (or vice versa), the structural interface is used as a possible interaction between CK and CKR. All intermolecular interactions were analyzed using Neighborhood and visualized using NGL[58].

### Epitopes
Complex structures containing CK or CKR components and non-CK or non-CKR components are distinguished using the UniProt ID from the metadata of the mmCIF file header as described in Section 2.1. Non-CK or non-CKR components in a complex structure include other binding partners from its cellular host, immunoglobulin, or nanobody. PDB chains with non-CK and non-CKR components in the CK or CKR protein structure complex possessing greater than 30% sequence similarity to a known B-cell antibody, nanobody, or T-cell receptor sequence are indexed in a dataset of antibodies. Structural complexes containing at

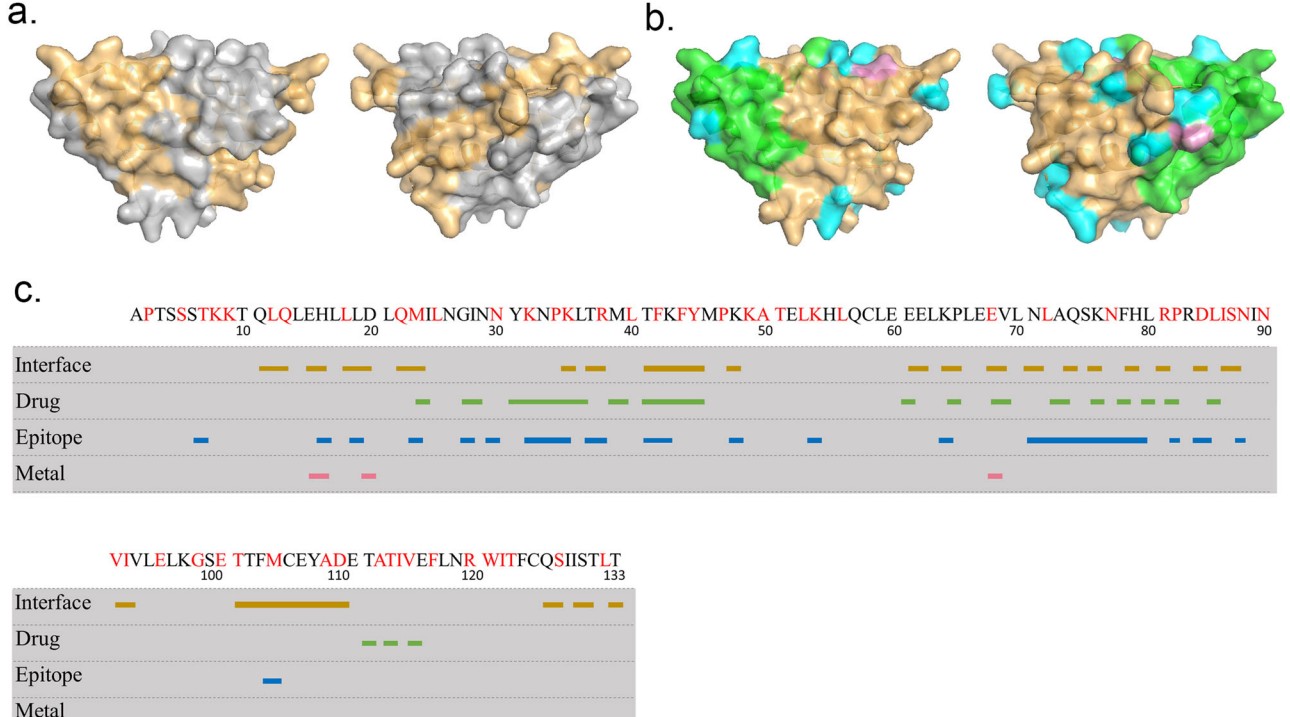

**Fig. 7 | Comprehensive structure in CytoSIP.** TNFA is used as an example for illustration. The longest segment of TNFA is selected as the representative structure for the alignment of all relevant interfaces to comparatively analyze the relative position of each interface on the structure. **a** Representative structure is colored in yellow, with the exception of its interaction interface colored in gray, the front and back view (horizontal flipped of 180 degrees) are shown; **b** Representative structure is colored in yellow, with drug surface colored in green, epitope colored in blue and metal binding sites colored in pink, the front and back view (horizontal flipped of 180 degrees) are shown; **c** TNFA sequence is shown alongside with interaction interface (gold strips), drug targeting surface (green strips), epitopes (blue strips) and metal binding sites (pink strips) (PDB ID: 1M4C).

least one antibody chain and at least one CK or CKR protein or glyco-protein chain are further analyzed using the Neighborhood database to identify all antibody-antigen interfaces. Epitopes interacting with a B-cell antibody or nanobody are annotated as B-cell epitopes, while epitopes interacting with T-cell receptors are annotated as T-cell epitopes. Epitopes defined by the antibody-antigen interface are also annotated using the source organisms used to produce antibodies.

Many CK or CKR antigen chains possess covalently linked glycosylation features assigned as individual glycan chains. These covalently linked glycan chains are considered an integral part of the CK or CKR protein as long as a characteristic covalent bond is identified for N- or O-glycosylation. Interfaces between the antibodies and the CK or CKR proteins or glycoproteins include mostly hydrophobic interactions and polar interactions such as hydrogen bonds and salt bridges. We characterize the conformational epitope of the CK or CKR protein as the collection of CK or CKR amino acids and glycan residues involved in the intermolecular interactions. The paratope is the collection of antibody amino acids and glycan residues located in the antibody-antigen interface.

### Small molecule and metal targeting surfaces
The intermolecular interactions between small molecules and CK/CKR proteins are stored as the hydrophobic interactions, hydrogen bonds, and other polar interactions between the small-molecule compound and the corresponding binding pocket. The polypeptide chain with the most extensive interactions with the small molecule is considered the CK or CKR source of the drug targeting surface. The chemical structure of each small molecule in the database is compared with that of the drug molecule reported in the DrugBank[59] using the open-source cheminformatics software RDKit[60]. The drug's approval status is annotated using information from the DrugBank as 'FDA approved,' 'in DrugBank,' or 'not in DrugBank'

for each small molecule. The corresponding generic drug name is also annotated for each compound in the DrugBank.

The intermolecular interactions between metal ions and CK/CKR proteins are stored as coordinating bonds and represent the constructed metal binding sites as metal targeting surface. Free metal ions with no significant coordinating bonds are removed from the dataset. The polypeptide chain that forms the most coordinating bonds with the subject metal ion is characterized and used as the CK or CKR source of the metal binding site. The quality of each metal binding site is evaluated using the previously described algorithm used in the validation of magnesium binding sites in nucleic acid structures[61], with modification in the validation of metal binding sites in viral structures[62].

### Representative structure
PDB structural chain with the highest similarity and completeness to a known CK or CKR sequence in the dataset is selected as the representative structure of the corresponding CK or CKR. For CK or CKR with no experimentally determined structures in the PDB, structural models predicted using Alphafold2[63] are used as representative structures to illustrate the fold. The representative structure of each CK or CKR is visualized using NGL, with options to map interaction interfaces, SNPs, or hotspots in different colors on the surface of the representative structure. The range of modeled representative PDB structure is mapped and annotated onto the full-length protein sequence from the UniProt. Topological and other site information, including Extracellular, Transmembrane, Cytoplasmic, Domain, and motif of each CK and CKR obtained from UniProt, are also mapped on the corresponding CK or CKR sequence for comparative annotation.

### Disease-associated genes and variant
Association data between CK/CKR genes and disease/variant are obtained using version 7.0 of the DisGeNET[64]. DisGeNET integrates data from public

databases, GWAS catalogs, animal models, and scientific literature, mainly containing disease and gene association (GDA) and variant and disease association (VDA). Diseases in the database are coded using UMLS® Concept Unique Identifiers (CUIs) and annotated with the UMLS® semantic type, the MeSH class, and the top-level concepts from the Human Disease Ontology and the Human Phenotype Ontology. The DisGeNET SQLite files contain data on 26137 genes, including all CK and CKR genes characterized in section 2.1. DisGeNET also contains information on genetic variants, including missense (28%), intronic (26%), frameshift and intergenic (both 11%) SNPs, while our database specifically stores only information related to the missense SNPs. All data are presented hierarchically by disease classification, specific disease, relevant CK and CKR genes, and their associated SNPs.

## Database and webserver implementation

All data are processed and organized into a PostgreSQL Database Management System. The backend of the cytoSIP database uses PostgreSQL 10.15 (database server). The web server is deployed using an Ubuntu Linux virtual machine running Nginx 1.14.0 and Gunicorn 20.0.4. The interface components of the website are designed and implemented using the Django template engine 3.1.4. Molecular graphics on the view page use HTML5 as implemented in the NGL Javascript library 2.2.1. Static molecular graphics used in the manuscript for demonstration purposes are produced using PyMOL 2.5.7 (Schrödinger), while visual art on the home page was purchased from https://www.veer.com/. cytoSIP has been tested in several popular web browsers, including Google Chrome 89.0.4389.82, Mozilla Firefox 87.0, Apple Safari 13.0.2, and Microsoft Edge 89.0.774.75. The styles of the web interface are optimized using the Bootstrap 4.5.0 library to accommodate both large computer screens and small screens on handheld devices. The cytoSIP webserver is accessible via https://cytosip.biocloud.top/.

## Data availability

All data supporting the findings of this study are available within the paper and its Supplementary Data 1. All other data are available from the corresponding author (or other sources, as applicable) on reasonable request.

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

## Acknowledgements

The authors would like to thank Mahendra D. Chordia and Can Zhang for valuable discussion and critical comments for the manuscript and the potential applications of the data. The work is funded by Natural Science Foundation of Hunan Province (2021JJ30101), National Natural Science Foundation of China (81970248, 81974019), the National Key Research and Development Program of China (2018YFA0108700), Guangdong Provincial Special Support Program for Prominent Talents (2021JC06Y656), and the Wisdom Accumulation and Talent Cultivation Project of the Third Xiangya Hospital of Central South University (YX202212). Nasui Wang is supported by the Science and Technology Planning Project of Guangdong Province of China (A2020198) and the Science and Technology Planning Project of Shantou of China (200620115260167).

## Author contributions

H.P.Z., P.Z., L.W., F.S., and M.Y.Z. conceived the study, designed the experiments, analyzed the data, and wrote the manuscript. HJ.M. made contributions to the acquisition and analysis, Q.Y.L., J.H.Z., H.H.Z., S.Y.C., H.W., and Y.M.Z. analyzed the data. Z.Q.X. and N.S.W. contributed to the editing of the manuscript.

## Competing interests

The authors declare no competing interests.
