## [Peer review file · Communications Biology]

Reviewers' comments:

Reviewer #1 (Remarks to the Author):

I think this is an interesting and useful resource for the community given the current interest in cytokines as therapeutics. The curation seems thorough, I hope the authors will continually update the website based on new data and user feedback.

Reviewer #2 (Remarks to the Author):

Communications Biology

Manuscript Number: COMMSBIO-23-1741-T

Corresponding Author: Professor Zheng

"CytoSIP: An annotated structural atlas for interactions involving cytokine or cytokine receptor", by Professor Zheng and colleagues.

The authors have recognised a major shortcoming in the field of cytokine therapeutics and have created a database to aid the development of agents targeting cytokine:cytokine receptor interactions. They have called their database, or atlas, CytoSIP – which is an apt description of the information which can be gleaned for specific cytokines or cytokine receptors, namely missense SNPs, interaction interface details and phenotypic information. The term cytokine has been used in the broadest sense and includes cytokines, growth factors, chemokines etc, making the CytoSIP database applicable to an extremely large number of proteins.

Although there are existing cytokine databases, and the authors cite some of these in their Introduction, they are very narrow in their focus and the data pipeline is often not integrated in a manner that allows users easy access. In contrast, CytoSIP is very easy to use and makes structural and disease information readily accessible. It should be noted that one of the existing cytokine databases has a very similar name to CytoSIP, it is the Cytokine Signaling (CytoSig) database at the NIH National Cancer Institute (<https://cytosig.ccr.cancer.gov/>). While the data/information provided by the two databases is quite different and does not overlap at all, the similarity in their names may cause some confusion in the cytokine research community and could potentially infringe on the intellectual property of the NIH National Cancer Institute, hopefully the authors have investigated this latter point?

The CytoSIP database will be of huge interest to the cytokine research community, however while testing out the website portal a number of issues and errors were identified and these need to be addressed by the authors. The errors/issues were generated using different browsers (ie. firefox and chrome), different operating systems (ie. MAC-OS Big Sur 11.7 and Windows 10) and different Windows 10 machines (ie. laptop and desktop computers).

(1) Cytokine receptor classification: for the examples that I tried these are not always correct and do not match the data from KEGG or UniProt. For example: (a) β -common receptor (IL3RB) should be classified as belonging to the "IL3 receptor family" but the assigned classification in CytoSIP is "IL8 receptor family"; (b) GM-CSF alpha receptor (CSF2R) should be classified as belonging to the "IL3 receptor family" but the assigned classification in CytoSIP is "IL7 receptor family"; (c) IL5 alpha receptor (IL5RA) should be classified as belonging to the "IL3 receptor family" but the assigned classification in CytoSIP is "IL4 receptor family"; (d) IL3 alpha receptor (IL3RA) is correctly assigned in CytoSIP.

The classification for the cytokines appears to be OK, at least for the numerous examples I tried.

(2) The representative structure that is displayed for the cytokine or cytokine receptor is not always the most complete structure that is available in the protein databank (PDB). For example, there are two entries in the PDB for the GM-CSF alpha receptor (CSF2R), ie. 4NKQ and 4RS1. The structure for the extracellular domain of GM-CSF alpha receptor (CSF2R) is (more or less) complete in 4RS1, whereas it is the very incomplete structure from 4NKQ that is displayed as the representative structure.

When there is no experimental structure for a cytokine or cytokine receptor, the AlphaFold2 model is consistently shown as the representative structure – so that seems to be working OK.

(3) An error is generated when trying to display metal binding sites: for all examples with a metal bound that I tried, I got an IndexError.

(4) An error is generated when trying to display drug binding sites: for all examples with a small molecule bound that I tried, I got a ProgrammingError.

(5) An unpopulated template window is generated when attempting to display the interaction interface from within the cytokine receptor menu: the interaction interface is displayed correctly if you use the cytokine menu but if you are in the cytokine receptor menu all you get is an unpopulated pop up window.

The SNP links to the NCBI SNP database, epitope displays and disease searching all worked seamlessly.

Overall the manuscript is well written and the description of the database elements is concise. Once all of the bugs identified in points 1-5 have been addressed, the CytoSIP database will be an extremely useful tool in the development of cytokine therapeutics.

Minor points to be addressed:

(6) The authors should state in the Methods section what software was used to generate the images in the figure panels, was it NGL?

(7) Figure legends should be standalone, ie. all details should be given in the legend itself. In most of the legends there are details missing, for example the name of the cytokine or receptor shown in the figure panel is missing. The authors should add the missing details into the legends:

- Fig. 1. CK and CKR should be used, rather than "cytokine" and "cytokine receptor" (same for the other figure legends). The specific CK and CKR shown in panel A should be stated, as should the CK in panels B and C. Also, it is customary to include the reference when citing a PDB structure for example (PDB ID: 3BQL, [ref number]).

- Fig. 2. It would be helpful for the different CKR to be labeled in each panel.

- Fig. 4. Identify the CK or CKR in panel A, it is not clear what the structure actually is – it looks like one protein is shown and not a complex? Panel C should be identified as the PPI interface of IL1 β :gevokizumab and in the legend IL1 β should be used and not IL-1beta. In panel C it is hard to read the residue labels with the surface displayed. There should be room to show two views of the interaction interface – one with the surface (as already shown) and one without the surface.

- Fig. 5. The legend should state that the results shown are for Acute Coronary Syndrome.

- Fig. 6. Web site should be one word, as "website" has been used elsewhere in the manuscript.

- Fig. 7. Are panels A and B for IL2 or TNFA? This should be stated in the legend.

(8) The abbreviation CK for "cytokine" and CKR for "cytokine receptor" is defined and used in the Abstract, Methods, Results and Discussion however in the Introduction there is a mixture of CK, CKR, cytokine and cytokine receptor. Also the format CK/CKR and CK-CKR are both used in the manuscript to indicate cytokine(s) in complex with their receptor(s). Only one format should be used throughout the manuscript.

(9) On the CytoSIP portal website there is a typographical error where CR/CKR is used instead of CK/CKR.

(10) Abbreviations should be defined when they are first encountered and the abbreviation used

thereafter, for example: (a) on line 66 G-CSF should be defined, ie "or granulocyte-colony stimulating factor (G-CSF) antibodies" and (b) on line 67, interleukin (IL) and interleukin receptor should be defined, ie "include antibodies against interleukin 6 receptor (IL6R) [16], IL9R [17]...".

(11) There should be consistency in the format of terms used in the manuscript, for example both "T cells" and "T-cells" have been used throughout the manuscript. Another example is the citing of figure panels in the main text, the majority have the panel indicated using an uppercase letter (ie Fig. 1A) but on line 255 the panels are indicated using lowercase letters (ie Fig. 3a-3c). Also on line 326 "Figure" is used instead of the abbreviation "Fig." to cite Fig. 2. Only one format should be used.

(12) A blank space is sometimes missing before a reference citation, for example on line 51 "patients[6]" should read as "patients [6]".

(13) On line 78, remove the fullstop in "d.iseases" so that it reads "diseases".

(14) Although CytoSIP is defined in the Abstract it is not defined in the Introduction on line 107.

(15) The reference for the NCBI SNP database is missing on line 142, this should be included.

(16) Fig. 4B should be cited on line 277 after "IL1 β :gevokizumab complex" so that it reads "IL1 β :gevokizumab complex (Fig. 4B)" and the citation of Fig. 4B on line 279 should be changed to Fig. 4C.

(17) On line 308, the authors specify "X-ray complex structures". The authors need to clarify if they are excluding CK/CKR complexes obtained using cryo-EM?

(18) Table 1 Legend. Use CK and CKR rather than "cytokine" and "cytokine receptor".

Something to consider:

While testing the database, it was often difficult to find the location of a specific SNP residue on the displayed structure, even when the surface representations were turned off and the structure enlarged. At a later date, the authors may want to consider adding the functionality of linking the residue location in the structure to the table entry, for example having the ability to "click" on the residue in the SNP table and the location of that residue is then highlighted in the structure or the structure moves to centre on the residue.

Reviewer #3 (Remarks to the Author):

The authors have developed a comprehensive database that focuses on cytokines and cytokine receptors, providing a wealth of useful information for the research community. I have a few minor suggestions:

1. Consider including a figure that illustrates the data collection sources and the processing steps involved. This visual representation will help readers grasp the overall design of the study more effectively.
2. It would be beneficial to include a supplementary table summarizing the cytokine/cytokine receptor families. This table can provide a concise overview of the different families and their associated members, aiding researchers in quickly accessing relevant information.
3. An interesting addition would be to explore the potential impact of single nucleotide polymorphisms (SNPs) on cytokine-cytokine receptor interactions. Algorithms like AlphaFold2 could be utilized to predict the effects of these genetic variations, providing valuable insights into the functional consequences of SNPs on cytokine signaling.
4. Consider addressing whether it is possible to export or download the search output from the

database's website. This functionality would enable users to save and analyze the retrieved data locally, facilitating further analysis and integration with other tools.

review from referee 1

I think this is an interesting and useful resource for the community given the current interest in cytokines as therapeutics. The curation seems thorough, I hope the authors will continually update the website based on new data and user feedback.

Thank you very much for the encouraging notes. We will put an effort into the continuous maintenance of the web resource.

review from referee 2

The authors have recognised a major shortcoming in the field of cytokine therapeutics and have created a database to aid the development of agents targeting cytokine: cytokine receptor interactions. They have called their database, or atlas, CytoSIP -which is an apt description of the information which can be gleaned for specific cytokines or cytokine receptors, namely missense SNPs, interaction interface details and phenotypic information. The term cytokine has been used in the broadest sense and includes cytokines, growth factors, chemokines etc, making the CytoSIP database applicable to an extremely large number of proteins.

We are much thankful for the summary outlining the objective and scope of our study. We hope that the resource will be helpful for the researchers in the broad field of cytokine biology.

Although there are existing cytokine databases, and the authors cite some of these in their Introduction, they are very narrow in their focus and the data pipeline is often not integrated in a manner that allows users easy access. In contrast, CytoSIP is very easy to use and makes structural and disease information readily accessible. It should be noted that one of the existing cytokine databases has a very similar name to CytoSIP, it is the Cytokine Signaling (CytoSig) database at the NIH National Cancer Institute(<https://cytosig.ccr.cancer.gov/>). While the data/information provided by the two databases is quite different and does not overlap at all, the similarity in their names may cause some confusion in the cytokine research community and could potentially infringe on the intellectual property of the NIH National Cancer Institute, hopefully the authors have investigated this latter point?

We have investigated the potential infringement of the database's intellectual properties. While there are hardly any legal resources or cases that deal with scientific database names, we believe we should stick to a higher standard and apply the rules of trademark infringement defined by the United States Patent and Trademark Office (USPTO) in our scenario. According to the USPTO definition, "Trademark infringement is the unauthorized use of a trademark or service mark on or in connection with goods and/or services in a manner

that is likely to cause confusion, deception, or mistake about the source of the goods and/or services." (<https://www.uspto.gov/page/about-trademark-infringement>). The similarity of services is defined by: 1) the intent to deceive; and 2) imitation in both name and logo design, both of which we took an additional step of consideration as described in more detail below.

While we did not intend to confuse or deceive the potential users, we have gone a step further by adding a banner on the CytoSIP home page as a disclaimer to avoid potential confusion about our database to the NIH CytoSig. The banner reads: "Disclaimer: This resource is NOT to be mistaken with the Cytokine Signaling (CytoSig) database (<https://cytosig.ccr.cancer.gov/>) at the NIH National Cancer Institute."

The first part of the database name "Cyto" actually refers to their similarity in the scientific field, which we believe it to be more beneficial than harmful. The second part of the database name does not use the same naming principle, provided that SIP is an acronym of three words all in uppercase, while Sig is the capitalized first three characters of a single word Signalling. To make the difference more profound, we have designed the CytoSIP logo to be completely dissimilar to the logo of the NIH CytoSig. We hope our due diligence should have freed us from the accusation of imitation in either name or logo design to cause confusion, deception, or mistake about the source of the services.

The CytoSIP database will be of huge interest to the cytokine research community, however while testing out the website portal a number of issues and errors were identified and these need to be addressed by the authors. The errors/issues were generated using different browsers (ie. firefox and chrome), different operating systems (ie. MAC-OS Big Sur 11.7 and Windows 10) and different Windows 10 machines (ie. laptop and desktop computers).

(1) Cytokine receptor classification: for the examples that I tried these are not always correct and do not match the data from KEGG or UniProt. For example: (a) β - common receptor (IL3RB) should be classified as belonging to the "IL3 receptor family" but the assigned classification in CytoSIP is "IL8 receptor family"; (b) GM-CSF alpha receptor (CSF2R) should be classified as belonging to the "IL3 receptor family" but the assigned classification in CytoSIP is "IL7 receptor family"; (c) IL5 alpha receptor (IL5RA) should be classified as belonging to the "IL3 receptor family" but the assigned classification in CytoSIP is "IL4 receptor family"; (d) IL3 alpha receptor (IL3RA) is correctly assigned in CytoSIP.

The classification for the cytokines appears to be OK, at least for the numerous examples I tried.

Thank you for pointing it out to us. We have carefully checked all cytokine and cytokine receptor classifications; the incorrect cytokine receptor classifications have been corrected.

(2) The representative structure that is displayed for the cytokine or cytokine receptor is not always the most complete structure that is available in the protein databank (PDB). For example, there are two entries in the PDB for the GM-CSF alpha receptor (CSF2R), ie. 4NKQ and 4RS1. The structure for the extracellular domain of GM-CSF alpha receptor (CSF2R) is (more or less) complete in 4RS1, whereas it is the very incomplete structure from 4NKQ that is displayed as the representative structure.

When there is no experimental structure for a cytokine or cytokine receptor, the AlphaFold2 model is consistently shown as the representative structure - so that seems to be working OK.

Thank you for raising the issue that representative structure is not always the most complete structure. We have revised the scoring criteria for selecting representational structures. It was fixed.

(3) An error is generated when trying to display metal binding sites: for all examples with a metal bound that I tried, I got an IndexError.

Thanks for your suggestion. It was fixed.

(4) An error is generated when trying to display drug binding sites: for all examples with a small molecule bound that I tried, I got a ProgrammingError.

The problem has been fixed.

(5) An unpopulated template window is generated when attempting to display the interaction interface from within the cytokine receptor menu: the interaction interface is displayed correctly if you use the cytokine menu but if you are in the cytokine receptor menu all you get is an unpopulated pop up window.

The problem has been fixed.

The SNP links to the NCBI SNP database, epitope displays and disease searching all worked seamlessly. Overall the manuscript is well written and the description of the database elements is concise. Once all of the bugs identified in points 1-5 have been addressed, the CytoSIP database will be an extremely useful tool in the development of cytokine therapeutics.

Thanks very much for pointing out the bugs in the database and the web server. We have gone through them one by one and debugged the code accordingly.

Minor points to be addressed:

(6) The authors should state in the Methods section what software was used to generate the images in the figure panels, was it NGL?

We have generated all the static images for demonstration purposes in the manuscript using PyMOL. We have added a new section "Database and web-server implementation" in Methods that contains details about the version of software applications we used through the manuscript and on the server.

(7) Figure legends should be standalone, ie. all details should be given in the legend itself. In most of the legends there are details missing, for example the name of the cytokine or receptor shown in the figure panel is missing. The authors should add the missing details into the legends:

·Fig. 1. CK and CKR should be used, rather than "cytokine" and "cytokine receptor" (same for the other figure legends). The specific CK and CKR shown in panel A should be stated, as should the CK in panels B and C. Also, it is customary to include the reference when citing a PDB structure for example (PDB ID:3BQL, [ref number]).

·Fig. 2. It would be helpful for the different CKR to be labeled in each panel.

·Fig. 4. Identify the CK or CKR in panel A, it is not clear what the structure actually is - it looks like one protein is shown and not a complex? Panel C should be identified as the PPI interface of IL1 β : gevokizumab and in the legend IL1 β should be used and not IL-1beta. In panel C it is hard to read the residue labels with the surface displayed. There should be room to show two views of the interaction interface - one with the surface (as already shown) and one without the surface.

·Fig. 5. The legend should state that the results shown are for Acute Coronary Syndrome.

·Fig. 6. Web site should be one word, as "website" has been used elsewhere in the manuscript.

·Fig. 7. Are panels A and B for IL2 or TNFA? This should be stated in the legend.

Thanks for your suggestion. We have rewritten all figure legends, aiming for clarity. Coloring schemes are mentioned in each figure legend to ensure the conformity of the standalone standard. The names of cytokines or cytokine receptors are annotated in each figure legend. We have also made corrections for comments towards individual figure legends:

Fig. 1: CK and CKR are used in the place of cytokine and cytokine receptor. Names of the specific CK or CKR are noted in parenthesis, followed by the PDB code of the structure used for illustration.

Fig. 2: The names of the CK and CKR are noted for each panel in the figure legends.

Fig. 4: We agree that the previous figure legend was ambiguous. We have rewritten the whole figure legend for Figure 4. Panel A indeed illustrates a single protein structure (TNFA), with SNPs found on the interaction interfaces marked in red. The legends for panel C have been updated according to the reviewer's comments, with IL1beta replaced by IL1 β , and labels for the residues with increased font size are shown above the surface.

Fig. 5: The figure legends state, "The results are shown for Acute Coronary Syndrome."

Fig. 6: The word website is now written in one word.

Fig. 7: The figure legend is now noted with "TNFA is used as an example for illustration."

(8) The abbreviation CK for "cytokine" and CKR for "cytokine receptor" is defined and used in the Abstract, Methods, Results and Discussion however in the Introduction there is a mixture of CK, CKR, cytokine and cytokine receptor. Also the format CK/CKR and CK-CKR are both used in the manuscript to indicate cytokine(s) in complex with their receptor(s). Only one format should be used throughout the manuscript.

Thanks for your suggestion. We have checked the consistent use of the CK and CKR abbreviations and updated them wherever appropriate. However, there are exceptions when the word cytokine is part of a phrase we did not abbreviate, e.g. cytokine storm, cytokine release syndrome, cytokine biology, etc. We have also corrected those places where CK/CKR was used to refer to the cytokine(s) in complex with their receptor(s) and made it CK-CKR consistently. For the abbreviation CK/CKR that referred to either CK or CKR and NOT their complex, we have spelled it out explicitly as "CK or CKR" instead of "CK/CKR" for clarity purposes, with a few exceptions when we need to use "CK/CKR" as a component in a larger context with other components, e.g. "Association data between CK/CKR genes and disease/variant ..."

(9) On the CytoSIP portal website there is a typographical error where CR/CKR is used instead of CK/CKR.

It has been fixed.

(10) Abbreviations should be defined when they are first encountered and the abbreviation used thereafter, for example: (a) on line 66 G-CSF should be defined, ie "or granulocyte-colony stimulating factor (G-CSF) antibodies" and (b) on line 67, interleukin (IL) and interleukin receptor should be defined, ie "include antibodies against interleukin 6 receptor (IL6R) [16], IL9R [17]...".

We have gone through the introduction section and ensured that all abbreviated CK or CKR acronyms are spelled out at their first encounter.

(11) There should be consistency in the format of terms used in the manuscript, for example both "T cells" and "T-cells" have been used throughout the manuscript. Another example is the citing of figure panels in the main text, the majority have the panel indicated using an uppercase letter (ie Fig.1A) but on line 255 the panels are indicated using lowercase letters (ie Fig. 3a-3c). Also on line 326 "Figure" is used instead of the abbreviation "Fig." to cite Fig. 2. Only one format should be used.

Thanks for noting the inconsistencies. We have updated the text to use "T-cell" consistently. We have also checked the reference to figures to use only the abbreviated "Fig." followed by the figure number. Whenever applicable, figure panels are ensured to be referenced using only uppercase letters.

(12) A blank space is sometimes missing before a reference citation, for example on line 51 "patients[6]" should read as "patients [6]".

A space is added prior to the reference number whenever applicable.

(13) On line 78, remove the fullstop in "d.iseases" so that it reads "diseases".

It has been fixed.

(14) Although CytoSIP is defined in the Abstract it is not defined in the Introduction on line 107.

The definition to CytoSIP was added to the last paragraph of the Introduction section.

(15) The reference for the NCBI SNP database is missing on line 142, this should be included.

The requested reference is added to the corresponding place.

(16) Fig. 4B should be cited on line 277 after "IL1 β :gevokizumab complex" so that it reads "IL1 β :gevokizumab complex (Fig. 4B)" and the citation of Fig. 4B on line 279 should be changed to Fig.4C.

Thank you for the thorough read. We agree with the proposed Figure reference and have updated the text exactly as the reviewer suggested.

(17) On line 308, the authors specify "X- ray complex structures". The authors need to clarify if they are excluding CK/CKR complexes obtained using cryo-EM?

CytoSIP catalogs all experimentally determined complex structures. We have updated the text accordingly. The text now reads, "These utilities allow for quick comparative analysis and visualization of interaction interfaces based on the experimentally determined complex structures."

(18) Table 1 Legend. Use CK and CKR rather than "cytokine" and "cytokine receptor".

According to comments (8), we have gone through the manuscript and updated all acronyms to CK or CKR

wherever applicable. This includes all figure legends and concurs with the proposed comment (18).

Something to consider:

While testing the database, it was often difficult to find the location of a specific SNP residue on the displayed structure, even when the surface representations were turned off and the structure enlarged. At a later date, the authors may want to consider adding the functionality of linking the residue location in the structure to the table entry, for example having the ability to "click" on the residue in the SNP table and the location of that residue is then highlighted in the structure or the structure moves to centre on the residue.

We fully appreciate the proposed feature and will implement it in the next release of cytoSIP.

review from referee 3

The authors have developed a comprehensive database that focuses on cytokines and cytokine receptors, providing a wealth of useful information for the research community.

Thanks very much for your positive comments!

I have a few minor suggestions:

1.Consider including a figure that illustrates the data collection sources and the processing steps involved. This visual representation will help readers grasp the overall design of the study more effectively.

We added text that annotates the data collection source to Figure 1. The processing steps involved were also amended in Figure 1 to enhance clarity.

2.It would be beneficial to include a supplementary table summarizing the cytokine/cytokine receptor families. This table can provide a concise overview of the different families and their associated members, aiding researchers in quickly accessing relevant information.

Thanks for your suggestion to improve the clarity of the data. We have included a supplementary table S1 summarizing the cytokine/cytokine receptor families.

3. An interesting addition would be to explore the potential impact of single nucleotide polymorphisms (SNPs) on cytokine-cytokine receptor interactions. Algorithms like AlphaFold2 could be utilized to predict the effects of these genetic variations, providing valuable insights into the functional consequences of SNPs on cytokine signaling.

We definitely agree that the proposed feature will improve the application scope of cytoSIP. However, this tool requires significant design and work and deserves a separate study. We will carefully consider crafting the proposed tool in the next release of cytoSIP.

4. Consider addressing whether it is possible to export or download the search output from the database's website. This functionality would enable users to save and analyze the retrieved data locally, facilitating further analysis and integration with other tools.

Thanks for your suggestion to improve the usability of the web service. We have added the functionality to download the search output from the database's website in CSV format.

Reviewers' comments:

Reviewer #2 (Remarks to the Author):

The authors have addressed most of the issues that were raised in the review of the original manuscript, however not all of the bugs have been (consistently) fixed.

Original point (1)

(1) Cytokine receptor classification: for the examples that I tried these are not always correct and do not match the data from KEGG or UniProt. For example: (a) β -common receptor (IL3RB) should be classified as belonging to the "IL3 receptor family" but the assigned classification in CytoSIP is "IL8 receptor family"; (b) GM-CSF alpha receptor (CSF2R) should be classified as belonging to the "IL3 receptor family" but the assigned classification in CytoSIP is "IL7 receptor family"; (c) IL5 alpha receptor (IL5RA) should be classified as belonging to the "IL3 receptor family" but the assigned classification in CytoSIP is "IL4 receptor family"; (d) IL3 alpha receptor (IL3RA) is correctly assigned in CytoSIP.

While the receptor classification is correct in the CytoKines Receptor search results table, if you then select a receptor entry in the results table to display the information for that specific receptor the classification is often still incorrect. Also, if you look at the results for specific CytoKines searches, the receptor classification is often incorrect. I have illustrated the four examples from original point (1) in a separate attachment.

The authors should try and sort out what is causing the incorrect receptor classification from being pulled into the results tables.

Original point (3)

(3) An error is generated when trying to display metal binding sites: for all examples with a metal bound that I tried, I got an IndexError.

For the many examples that I tried the metal binding site is now displayed nicely (for example IL1A). However, there are still instances where you get IndexError, for example IL21R (PDB ID: 4NZD) gives IndexError.

The authors should try and sort out what is causing the IndexError (if possible).

New bug/issue:

(1) I noticed that when looking at the results for a particular cytokine or cytokine receptor, the structure is displayed under the "Protein Structure" tab but the "Info" tab only shows an empty table. I have illustrated this problem for CSF2R in the separate attachment.

Minor points to be addressed:

(2) For any software used, the specific version should be stated in the Methods section where possible, for example PyMOL version 2.1.1.

(3) In Discussion section 4.2, "PDBid" has been used on the third line. This should be changed to "PDB ID".

Overall the revised manuscript is well written and the majority of the bugs/issues raised previously have been addressed. The authors should sort out the bugs/issues described above prior to the publication of the manuscript. If the ability to report a bug or problem was incorporated into the CytoSIP website, this would assist the authors in trouble shooting and maintaining the integrity database.

Reviewer #3 (Remarks to the Author):

The authors addressed most of my comments. In their rebuttal letter, they mentioned they added data collection source and processing step to Figure 1. But I didn't see the changes in Figure 1, is it missed somewhere? If they can fix this minor issue, I would agree to accept the manuscript.

Reviewers' comments:

Reviewer #2 (Remarks to the Author):

The authors have addressed most of the issues that were raised in the review of the original manuscript, however not all of the bugs have been (consistently) fixed.

Original point (1)

(1) Cytokine receptor classification: for the examples that I tried these are not always correct and do not match the data from KEGG or UniProt. For example: (a) β -common receptor (IL3RB) should be classified as belonging to the “IL3 receptor family” but the assigned classification in CytoSIP is “IL8 receptor family”; (b) GM-CSF alpha receptor (CSF2R) should be classified as belonging to the “IL3 receptor family” but the assigned classification in CytoSIP is “IL7 receptor family”; (c) IL5 alpha receptor (IL5RA) should be classified as belonging to the “IL3 receptor family” but the assigned classification in CytoSIP is “IL4 receptor family”; (d) IL3 alpha receptor (IL3RA) is correctly assigned in CytoSIP.

While the receptor classification is correct in the CytoKines Receptor search results table, if you then select a receptor entry in the results table to display the information for that specific receptor the classification is often still incorrect. Also, if you look at the results for specific CytoKines searches, the receptor classification is often incorrect. I have illustrated the four examples from original point (1) in a separate attachment.

The authors should try and sort out what is causing the incorrect receptor classification from being pulled into the results tables.

We would like to sincerely thank Reviewer #2 for pointing out this bug and providing the corresponding examples that allow us to debug the original problem systematically. The misclassification of some CKRs was due to the presence of incorrect redundant data in an underlying data table. We have now thoroughly checked this table to remove the redundancy and fetch the classification information directly from the original table. After the update, we checked the four examples raised by the reviewer; all receptor classifications are now correct and are consistent with data in the CytoKines Receptor search results table. The URLs for these examples is provided for convenience herein:

https://cytosip.biocloud.top/ckr_chains/75

https://cytosip.biocloud.top/ckr_chains/20

https://cytosip.biocloud.top/ckr_chains/77

Original point (3)

(3) An error is generated when trying to display metal binding sites: for all examples with a metal bound that I tried, I got an `IndexError`.

For the many examples that I tried the metal binding site is now displayed nicely (for example

IL1A). However, there are still instances where you get IndexError, for example IL21R (PDB ID: 4NZD) gives IndexError.

The authors should try and sort out what is causing the IndexError (if possible).

We have debugged the potential cause of the IndexError. IL21R (PDB ID: 4NZD) now shows the expected metal correctly at the following URL: <https://cytosip.biocloud.top/ngl/4nzd/A307-NA/ck>

New bug/issue:

(1) I noticed that when looking at the results for a particular cytokine or cytokine receptor, the structure is displayed under the “Protein Structure” tab but the “Info” tab only shows an empty table. I have illustrated this problem for CSF2R in the separate attachment.

We have fixed this error. The “Info” tab now shows “4rs1, Granulocyte-Macrophage Colony-Stimulating Factor Receptor Subunit Alpha” for the particular protein CSF2R mentioned by the reviewer.

Minor points to be addressed:

(2) For any software used, the specific version should be stated in the Methods section where possible, for example PyMOL version 2.1.1.

Thank you for spotting the omission. We have added the version number for PyMOL and NGL javascript library, which was not previously specified.

(3) In Discussion section 4.2, “PDBid” has been used on the third line. This should be changed to “PDB ID”.

We have changed the “PDBid” to “PDB ID” at the place. We have also searched through the manuscript to ensure that place involving the phrase “PDB ID” are spelled consistently.

Overall the revised manuscript is well written and the majority of the bugs/issues raised previously have been addressed. The authors should sort out the bugs/issues described above prior to the publication of the manuscript. If the ability to report a bug or problem was incorporated into the CytoSIP website, this would assist the authors in trouble shooting and maintaining the integrity database.

Thank you for the encouragement and for pointing out the bugs that had slipped our previous examination, which has resulted in a much-improved website. We have sorted out the bugs/issues described above one by one. We hope that the current version of the manuscript addresses all the issues that have been raised so far.

Reviewer #3 (Remarks to the Author):

The authors addressed most of my comments. In their rebuttal letter, they mentioned they added data collection source and processing step to Figure 1. But I didn't see the changes in Figure 1, is it missed somewhere? If they can fix this minor issue, I would agree to accept the manuscript.

Thank you for pointing out the mistake in the description of our previous rebuttal letter. Upon checking further, we noticed that the data collection sources and processing steps were added to Figure 6 and not Figure 1. Sorry for the confusion that we had carelessly caused in composing the response in the previous rebuttal letter.

REVIEWERS' COMMENTS:

Reviewer #2 (Remarks to the Author):

The authors have addressed all of the issues that were raised in the review of the original and revised manuscript.

The CytoSIP structural atlas portal is easy to use and will be an extremely useful resource for cytokine research.